# THz MEMS Switch Design

**DOI:** 10.3390/mi13050745

**Published:** 2022-05-08

**Authors:** Yukang Feng, Han-yu Tsao, N. Scott Barker

**Affiliations:** Department of Electrical and Computer Engineering, University of Virginia, Charlottesville, VA 22904, USA; ht5fy@virginia.edu

**Keywords:** millimeter-wave, terahertz, MEMS, switch, transmission line model

## Abstract

In this work, an mm-wave/THz MEMS switch design process is presented. The challenges and solutions associated with the switch electrical design, modeling, fabrication, and test are explored and discussed. To investigate the feasibility of this design process, the switches are designed on both silicon and fused quartz substrate and then tested in the 140–750 GHz frequency range. The measurement fits design expectations and simulation well. At 750 GHz the measurement results from switches on both substrates have an ON state insertion loss of less than 3 dB and an OFF state isolation larger than 12 dB.

## 1. Introduction

Compared with conventional diode-based RF switches, MEMS switches have significant advantages in RF performance including higher isolation, lower insertion loss, and fewer intermodulation products. Meanwhile, since the MEMS switch does not require constant DC bias current in the static ON and OFF states, it consumes nearly zero power [1]. Because of these advantages, significant effort has been made to develop MEMS switches in the centimeter band [2,3,4,5,6], and some had also been successfully introduced in commercial applications [7,8]. In comparison, MEMS switch development in the millimeter-wave or THz spectrum faces more challenges.

In such frequencies, the switch’s physical dimensions are comparable with its RF signal wavelength. Instead of regarding the switch as a lumped element, one needs to model the circuit from a transmission line perspective. The EM finite analysis is also frequently applied in RF optimization. Meanwhile, at such high frequencies, the switch RF performance improvement requires adjusting the circuit’s physical features in dimensions of micro-meters. Inevitably, the electrical design has to trade off with the fabrication techniques and limitations; which makes mm-wave/THz MEMS switch design more challenging.

In 2010, a DC contact MEMS switch operating at 50–100 GHz was reported [9]. Its center conductor in the coplanar waveguide (CPW) was actuated through a long cantilever beam from one side of the switch. The cantilever was driven by a comb-electrode actuated folded spring structure, which significantly complicated the bias structure and increased the circuit size. Another CMOS-based MEMS switch was successfully demonstrated at 220 GHz [10]. In this design, the air bridge structure that supported the actuator introduced a large parasitic capacitance and limited the isolation performance. Two switches had to be placed in series to provide 12 dB isolation. The switch reported in 2017 [11] successfully integrated a MEMS switch with a BiCMOS process. In the 220–320 GHz band this capacitive switch achieved 1 dB insertion loss and 12 dB isolation. Another 500–750 GHz waveguide switch [12] reported in 2017 used a MEMS-reconfigurable surface to block & unblock the wave propagation through the waveguide; however, this structure is incompatible with planar RF circuits.

In our previous work [13], we demonstrated an RF MEMS switch on a silicon substrate and provided the preliminary measurement in the 500–750 GHz (WR1.5) band. To investigate the feasibility of this design process on different dielectric substrates in a wider millimeter/THz spectrum, the switches were designed on both high resistivity silicon and fused quartz substrate and then tested in the 140–750 GHz frequency range. In this work, we provide a method to design RF MEMS switches in the mm-wave/THz frequency band. The challenges associated with the switch electrical design, modeling, fabrication, and test are discussed and the solutions are provided.

## 2. Mechanical Design

In this work, we focus on developing a series switch using a CPW structure as shown in Figure 1. Under applied external bias voltage, the electrostatic force pulls the cantilever towards the bias pad. The voltage that provides sufficient force that can turn the switch to an ON state is the actuation voltage. Once the external bias voltage reduces, the cantilever starts to restore to its initial position and returns to the OFF state under zero bias. In Figure 2, the switch’s ON/OFF states are presented.

The first step of this switch design is to choose the desired substrate. Among several widely used materials, high resistivity silicon and fused quartz are chosen. These two substrates have significant differences in physical features that lead to different approaches in their mechanical and electrical designs. Silicon, with its higher relative permittivity ($\epsilon_{r}=11.9$) than fused quartz ($\epsilon_{r}=3.8$), takes less physical length to build the same RF circuit on a silicon substrate, moreover, the transmission line features are also narrower. This can be an advantage to develop a more compact device but requires finer adjustment on the device RF optimization. On the other hand, depending on the type of fused quartz been used, it could have a dielectric strength as high as 10 MV/cm [14] in comparison with 0.3 MV/cm for resistivity silicon substrates. Therefore, using equivalent circuit dimensions, fused quartz substrate can potentially support 30× higher DC bias voltage than the silicon substrate. Such an additional safe margin provides more design flexibility for designing electrostatic actuated MEMS devices. Meanwhile, high resistivity silicon’s resistivity is at the order of $10^{4}$ Ω·cm [15]; in comparison, fused quartz has a higher resistivity at the order of $10^{14}$–$10^{16}$ Ω·cm [16]. A thin layer of silicon dioxide should be deposited on top of the silicon surface to improve DC isolation.

The actuator of the proposed RF MEMS switch can be modeled as a cantilever from a mechanical perspective. The cantilever’s spring constant *k* can be calculated by the following equation [17]:(1)$$k=2Ew{\left(\frac{t}{l}\right)}^{3}\frac{1-(\frac{x}{l})}{3-4{\left(\frac{t}{l}\right)}^{3}+4{\left(\frac{t}{l}\right)}^{4}}.$$

Its pull-down voltage $V_{pull}$ can be estimated by the following equation [1]:(2)$$V_{pull}=\sqrt[{}]{\frac{8kg^{3}}{27\epsilon_{0}L_{1}w}}.$$

Here, *E* is Young’s modulus of the cantilever material; *x* is the distance from the cantilever anchor to the center of the bias pad; *l*, *t*, and *w* are the cantilever’s length, thickness, and width. As shown in Figure 1 and Figure 2, *g* and $L_{1}$ are the actuation gap and the length of the actuation pad beneath the actuator.

A small dimple structure is attached beneath the cantilever tip. Its thickness *h* is 0.2 μm. This dimple reduces the contact surface at the switch’s ON state to reduce the Van der Waals forces.

Previous research [17] included a similar sized dimple beneath the cantilever beam tip, and the experimental result suggests a restoring force of 0.07 mN is needed for reliable restoration. Assuming gold as the cantilever material, *l* in a range of 70–100 μm and *x* is roughly half of *l*, *w* and *t* can be calculated accordingly. In consideration of fabrication feasibility, *w* should be at least 2–3 times longer than *t*. With an actuation gap *g* of 1.2 μm, the initial values of *t* and *w* are selected as 2.3 μm and 7 μm, respectively. The resulting $V_{pull}$ is estimated to be around 55 V.

To prevent dielectric breakdown during actuation, the electric field between the bias pad and nearby CPW should be kept well below the substrate dielectric strength. As presented in Figure 3, a crude estimate of the electric field strength can be obtained by assuming a uniform field distribution.

In the fused quartz-based design, the strongest electric field extends along the minimum gap between the CPW and adjacent bias pad. Under a 55-V bias voltage, the electric field is approximately 0.11 MV/cm along the 5 μm gap. This value is well below the substrate’s dielectric strength and can be further reduced by increasing the gap between the bias pad and CPW. By comparison, in the silicon based design, the electric field extended through the silicon dioxide thickness as depicted in Figure 3. Considering silicon dioxide’s dielectric strength of around 3–5 MV/cm, the silicon dioxide insulation layer should be around 100 nm to prevent dielectric breakdown. One common method to deposit a silicon dioxide layer is through plasma-enhanced chemical vapor deposition (PECVD). However, previous fabrication and subsequent tests suggest this PECVD oxide has a high risk of creating an un-covered silicon area known as pinholes inside the oxidation layer that result in potential shorts between the MEMS devices and silicon substrate [18,19]. In this work, a very uniform dry thermal silicon-dioxide layer was grown on the high resistivity silicon surface with a thickness of 100–110 nm.

## 3. Electrical Design

In Figure 4, a cross-section view of a coplanar waveguide (CPW) structure is presented. The impedance of a CPW is largely determined by the signal line’s width, *w*, and the signal to ground gap, $g_{c}$. When scaling *w* and $g_{c}$ by the same ratio, CPW’s impedance has little variation. At a minimum feature size $g_{c}$ = 4 μm, the corresponding *w* to realize a 50 Ω impedance on different substrates is simulated through Ansys HFSS and listed in Table 1. The result shows that a lower relative permittivity substrate require a large signal line width. On silicon, the required signal line width is 7 μm, while for a fused quartz, the signal line width increases to 35 μm.

At the OFF state, the switch’s actuator is coupled to the bias pad and the CPW signal line through parasitic capacitances as presented in Figure 5. Here $C_{1}$ is actuator-bias pad coupling capacitance; $C_{2}$ is pad-CPW coupling capacitance; $C_{tip}$ is the coupling capacitance between actuator tip and CPW. The equivalent series capacitance of $C_{1}$ and $C_{2}$ is designed to be much smaller than $C_{tip}$ so that the total parasitic capacitance $C_{total}$ ≈ $C_{tip}$. At the OFF state, the switch’s RF isolation is determined by the impedance of the total capacitance.

To reach a higher RF isolation, a smaller $C_{tip}$ is desired. Assuming the switch actuator and CPW overlaps by 2 μm, the initial switch designs on silicon and fused quartz are each shown in Figure 6a and Figure 7a. Using the parallel plate capacitance equation, the tip capacitance of the silicon and quartz designs are estimated to be 0.6 fF and 2 fF, respectively. From these initial capacitance values, their isolation performance is simulated through a simplified transmission line circuit model. The results are plotted in blue and black curves in Figure 8. In this work, limited by equipment availability, the highest measurement frequency is 750 GHz; so the simulation is plotted in the DC-750 GHz frequency range. For the purposes of switch design comparison, a 15 dB isolation level is drawn as a reference. The estimation suggests the silicon design will have 15 dB isolation at around 470 GHz; due to the wider cantilever causing larger capacitance, the isolation of the quartz design is reduced to 15 dB at about 140 GHz.

To improve the isolation performance, the overlapping area that forms $C_{tip}$ must be trimmed to minimize the cantilever’s parasitic capacitance. In the silicon switch, as shown in Figure 6b, the cantilever tip and the CPW’s signal line tip are both tapered from 7 μm to 3.5 μm. With a smaller overlapping area, the capacitance is reduced to 0.3 fF. In the quartz switch, the cantilever width is reduced to 8 μm. To match the elevated CPW impedance to 50 Ω, the horizontal gap between signal and ground is adjusted to 15 μm; the CPW’s signal line tip is also tapered to 3 μm. This adjustment is presented in Figure 7b. Through trimming the geometries of the cantilever and the CPW, the parasitic capacitance of the quartz switch is significantly reduced to roughly 0.5 fF. The estimated RF isolation of both silicon and fused quartz switches are plotted in green and red in Figure 8. The fused quartz switch’s cut-off frequency with 15 dB isolation is expanded to 560 GHz, in comparison, the silicon switch’s 15 dB isolation bandwidth is higher than 750 GHz.

At the ON state, the switch’s RF performance can also be analyzed through a transmission line model. As presented in Figure 1 and Figure 2, the actuator in the MEMS switch creates a discontinuity in physical geometry and impedance mismatch. The switch’s actuator and its adjacent ground lines can be modeled as a CPW section, which is elevated from the substrate. As the CPW elevates further away from the silicon substrate surface, the effective relative permittivity $\varepsilon_{eff}$ for the CPW decreases and the impedance increases. This relationship is presented in Figure 9a. In Table 2, the simulated CPW impedance with different elevation heights above the silicon substrate is provided.

Because the actuator beam is elevated to 1.2 μm above the substrate, the effective relative permittivity $\varepsilon_{eff}$ reduces to around 4.8, the impedance rises to 70 Ω, and the return loss will be more than 10 dB at 700 GHz. Such RF performance change can be explained through a simplified transmission line model in Figure 10. For example, at 3 GHz, a 70 μm cantilever beam has an electrical length of merely 0.9${}^{\circ }$ and the resulting impedance mismatch is negligible. In comparison, at 300 GHz, the electrical length significantly increases to 90${}^{\circ }$, which naturally has a significant impact on the RF performance.

The cantilever’s impedance mismatch is most significant at the frequency when it reaches 90${}^{\circ }$ electric length. Here $Z_{c}$ is the introduced resistance due to switch Ohmic contact, on a scale of around 1 Ω or less. The $Z_{a}$ section is a quarter wavelength long and therefore, the input impedance at is given by:$$Z_{in}=Z_{a}^{2}/Z_{0}$$

In this case, the equivalent impedance of $Z_{in}$ is 98 Ω.

In order to reduce this impedance mismatch, the DC bias pad size is engineered to reduce the actuator impedance $Z_{a}$. Since the coplanar waveguide’s characteristic impedance is approximated as:(3)$$Z=\sqrt{\frac{L}{C}},$$

by increasing the bias pad’s size, the total parasitic capacitance at the actuator area is doubled, and the impedance is then successfully tuned to 48 Ω. This engineering tuning method is presented in Figure 9b.

A more comprehensive circuit model includes the anchor and the actuator’s tip is provided in Figure 11 and Figure 12 respectively. Figure 11 represents the ON state, in which the actuator is pulled down to make tip contact at the free end. $Z_{anc}$ represents the anchor section of the actuator; $Z_{a}$ represents the elevated actuator after impedance tuning. $Z_{elev}$ represents the elevated CPW section assuming no fringing capacitance impact from the bias pad. The switch’s transmission line model for the OFF state is shown in Figure 12. The gap beneath the actuator tip serves as an isolation capacitor and introduces impedance $Z_{c1}$.

The simulation result of a silicon-based switch at the ON state is presented in Figure 13a. In this figure, both the transmission line model and HFSS finite element analysis are included and compared. The insertion loss is smaller than 1 dB across DC-750 GHz band, while the return loss is better than 15 dB. Compared with EM simulation, the transmission line model catches the major scale and trends of insertion loss and return loss.

The silicon switch’s OFF state simulation is given in Figure 13b. The result shows the expected OFF state isolation better than 15 dB. The transmission line model fits the EM simulation result well in the isolation plots (S21/S12). In the reflection plots (S11/S22), the transmission line model does not account for the radiated energy. As a result, its simulation result has a minor difference from HFSS analysis. Similar results are also seen in fused quartz based simulation in Figure 13c,d.

## 4. Switch Fabrication Challenge & Solution

The silicon-based MEMS switch fabrication flow is simplified and presented in Figure 14. The fabrication process starts with circuit layer deposition using the lift-off technique. After that, two aluminum sacrificial layers are deposited and the dimple position is prepared with another lift-off. Dry etching and plating were used to form the anchor of switch. After the gold seed layer and photoresist patterning, the actuator beam was plated. After a series of wet etch steps, the switch is finally released by a critical point dryer (CPD). The quartz-based MEMS switch fabrication process is very similar.

Typical MEMS fabrication uses photoresists such as SU-8 as a sacrificial layer, which has a coefficient of thermal expansion (CTE) as high as 50–102 ppm/K [20,21]. In comparison, Aluminum’s CTE is 25.5 ppm/K, much closer to Gold’s CTE of 13.9 ppm/K. In this research, using an Al sacrificial layer significantly reduced the CTE mismatch between the actuator beam and the sacrificial layer beneath it, which prevented the commonly observed beam bowing after device release [22].

The Al sacrificial layer can also be easily removed through halogen gas-based reactive ion etch (RIE) or alkali solution wet etch. However, using an Al sacrificial layer also brings challenges in fabrication because it can easily form Al-Au compounds which cannot be easily removed. This red Al-Au compound can be observed in Figure 15a. A chromium layer is added as a diffusion barrier layer between the Al sacrificial layer and Au circuit layer [23,24]. In Figure 14d–e, a chlorine RIE and wet etch combined procedure is used to remove the aluminum layer and the chromium barrier layer.

## 5. Switch Calibration & Measurement

In order to verify the switch design, the RF MEMS switch is fabricated and tested. An on-wafer measurement is set up as presented in Figure 16. A Keysight VNA (PNA-5245A) is used to conduct a two-port S-parameter measurement. Four sets of VNA extender pairs (VDI WR-5.1, WR3.4, WR2.2, and WR-1.5) are used to up-convert VNA test frequency sweeping to 140–220 GHz, 220–330 GHz, 330–500 GHz, and 500–750 GHz. To conduct the on-wafer measurement, DMPI T-wave probes (WR5.1, WR3.4, WR2.2, and WR1.5) are also used correspondingly [25]. An SEM image of the calibration kit and completed RF MEMS contact switch is provided in Figure 17. The switch’s DC high voltage bias and ground are provided through a bias tee integrated into the DMPI probes. A series 10 MΩ resistance is placed in the DC bias circuit to limit the current below 10 μA in the potential breakdown condition.

Before measurement, on-wafer calibration is applied using Through-Reflection-Line (TRL) standards and WinCal XE calibration software. Standards include a 120 μm through, two reflection and three lines. The three different lines have additional lengths of 45 μm, 67 μm and 113 μm. Redundant reflections and lines are used for higher calibration accuracy [25]. Calibrated measurement shows the loss of 50 Ω CPW is around 3.5 dB/mm at 500 GHz and around 6.5 dB/mm at 750 GHz.

The silicon switch measurement is compared with corresponding HFSS simulation results in Figure 18a,b. The ON state performance is shown in Figure 18a. The measurement shows the switch’s impedance is well matched and has a small variation over frequency. The return loss is better than 10 dB in the whole band. The switch’s insertion loss is thus dominated by the actuator contact resistance and metal loss. Previous discussion demonstrated the contact resistance will bring insertion loss merely on a scale of 0.1 dB across the whole band; meanwhile, the metal loss will cause switch insertion loss to increase slightly over frequency. The measured insertion loss (S21 and S12) in Figure 18a fits the expectation. The return loss (S11 and S22) is mostly better than 20 dB as HFSS simulation suggested. Because the measurement is conducted in four different waveguide bands through four calibrations, there are inevitably certain minor measurement errors at the boundary frequency points. However, the return loss measurement trend still fits the simulated curve.

The silicon switch’s OFF state measurement is presented in Figure 18b. Because the CPW sections are isolated by the air-gap capacitance, any incident RF energy is mostly reflected. In this measurement, the return loss (S11 and S22) curves are only slightly lower than 0 dB, which fits the simulated result well. The isolation plot (S12 and S21) also match the simulated curves.

A similar condition is also observed in the fused quartz switch’s ON and OFF state measurements. Comparing Figure 13a,c, the silicon switch’s impedance match is expected to be better than the fused quartz switch in the 200–500 GHz band, which is due to different bias pad-actuator coupling conditions and associated resonance. The measurement in Figure 18a,c matches such expectation. Meanwhile, the fused quartz switch and silicon switch have similar insertion loss conditions, which is very reasonable. In both switch designs, insertion loss is majorly determined by the actuator’s contact resistance and gold metal loss. In both switches, the actuation bias voltage is 55 V, and both actuators have the same thickness. The actuation force is expected to be similar, which leads to the same expected contact resistance. Gold is used in both switch designs, which provides the same metal loss over frequency. Those factors determine both switches to have fairly similar and constant insertion loss over frequency; as frequency increases, the insertion loss increases by a similar magnitude in both silicon and fused quartz designs.

The quartz switch measurement in Figure 18d also follows the HFSS simulation result well. Similar to the silicon switch’s OFF state measurement, the fused quartz switch has 0.3–0.4 fF air gap capacitance. The impedance of this capacitance is dominant compared with 50 Ω transmission line impedance. As frequency increases, the reduced capacitance impedance leads to reduced isolation and rising S21/S12 measurement curves over frequency.

## 6. Discussion & Conclusions

In this paper, a THz MEMS switch design process is presented. To validate this process, THz MEMS switches are realized on both silicon and fused quartz as examples of both high and low dielectric constant substrates respectively.

In the device design, electrostatic actuation is selected to control the switches in consideration of its advantage over device size, integration challenges, switching speed, power consumption, and RF performance. Manufacturing limits and mechanical reliability together determined the minimum dimensions and thus the switches’ geometries. To integrate elevated actuators with CPW, the bias pad geometry is engineered to provide an optimized impedance match. The designed MEMS switches are modeled through transmission line analysis as well as finite element-based electromagnetic simulations. The comparison suggests the transmission line model captures the major electrical features successfully; meanwhile, the finite element model can also evaluate certain minor coupling, resonance, and frequency-dependent conductor loss. Fabrication flow of the THz MEMS switch is provided, in which a diffusion barrier layer is used to prevent forming Au-Al compound [26]. A new RIE/wet-etch combined process is critical to selectively etch certain metal layers. Both silicon and fused quartz switches are calibrated through the two-port TRL method from 140 to 750 GHz. The measurements fit previous modeling and simulation results well and serve to verify this THz MEMS switch design process.

## Figures and Tables

**Figure 1 micromachines-13-00745-f001:**
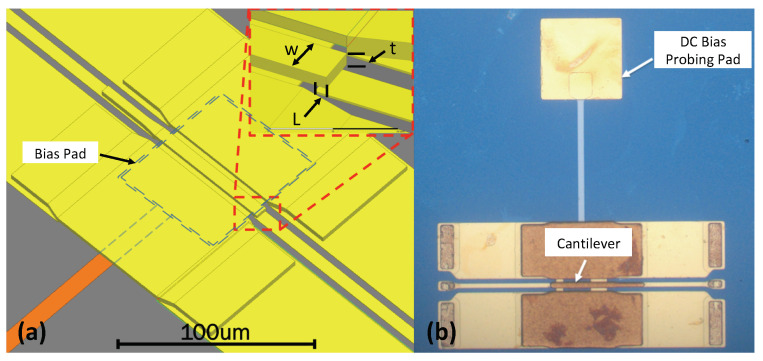
(**a**) 3-D model of an electro-static actuated MEMS switch in a series configuration. The switch’s OFF state isolation is provided an air gap between the actuator’s tip and the CPW section’s tip. When applying external DC bias, the electrostatic force between the bias pad and the actuator will pull the actuator down and provide an RF signal path. (**b**) A photomicrograph of a fabricated switch.

**Figure 2 micromachines-13-00745-f002:**
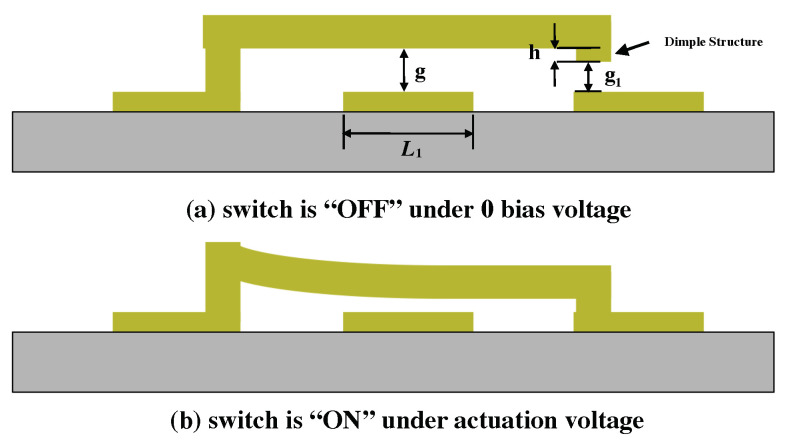
The OFF and ON states of the RF MEMS switch under different DC bias conditions. The cantilever to bias pad gap *g* and dimple to CPW distance $g_{1}$ are both marked. (**a**) The Switch is under OFF state. (**b**) The Switch is under ON state.

**Figure 3 micromachines-13-00745-f003:**
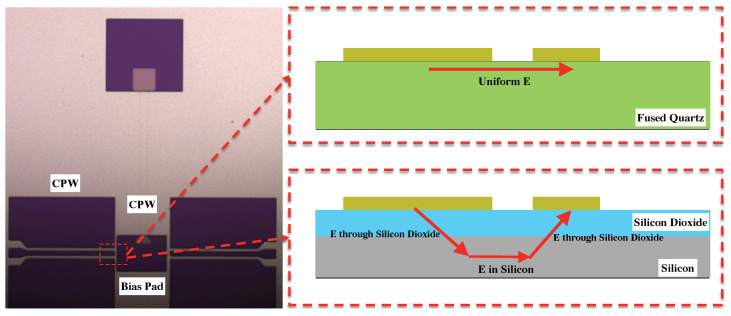
The strongest electric field is between the CPW and the adjacent DC bias pad.

**Figure 4 micromachines-13-00745-f004:**
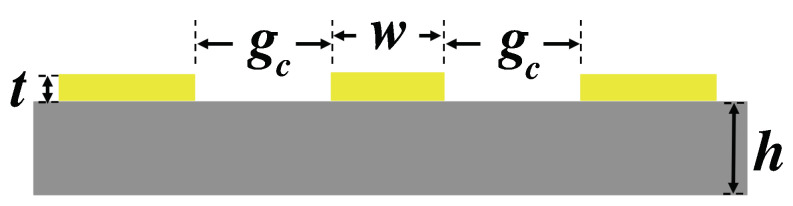
The cross-section view of a CPW. In this work, both high resistivity silicon and fused quartz substrate have thickness h of 500 μm. The CPW is evaporated gold with a thickness t of 400 nm. The signal line width *w* and signal-ground gap *g* are impacted by the substrate’s relative permittivity.

**Figure 5 micromachines-13-00745-f005:**
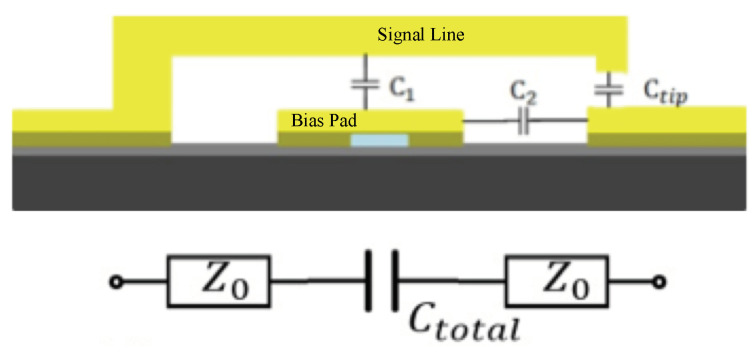
The parasitic capacitance of MEMS switch under OFF state and the switch’s equivalent circuit model.

**Figure 6 micromachines-13-00745-f006:**
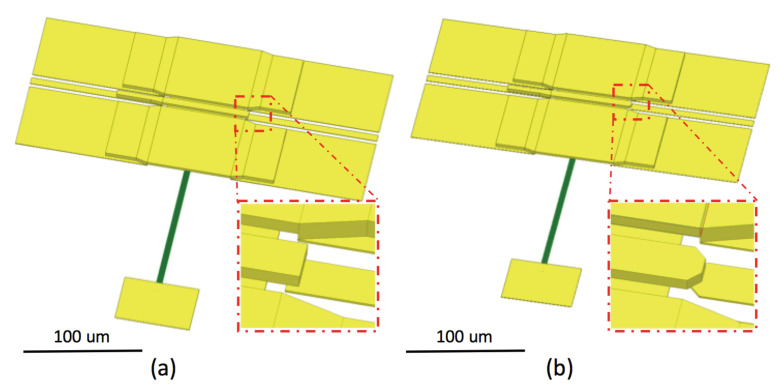
Three dimensional (3-D) model of high resistivity silicon-based MEMS switch. The initial design (**a**) has a larger parallel area at the actuator tip, which causes larger capacitance. To reduce such capacitance for higher isolation performance, its actuator tip is trimmed (**b**).

**Figure 7 micromachines-13-00745-f007:**
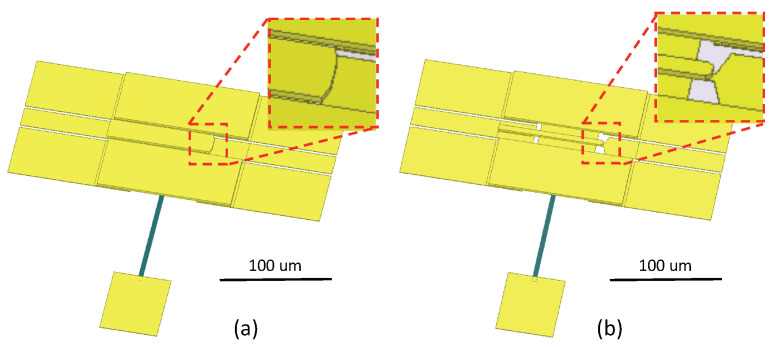
Three dimensional (3-D) model of fused quartz-based MEMS switch. In the initial design (**a**) the actuator has the same width as the CPW’s signal line, which causes roughly 2 fF parasitic capacitance at the tip. To reduce such capacitance, the actuator’s width was reduced and the tip is trimmed to further reduce the capacitance (**b**).

**Figure 8 micromachines-13-00745-f008:**
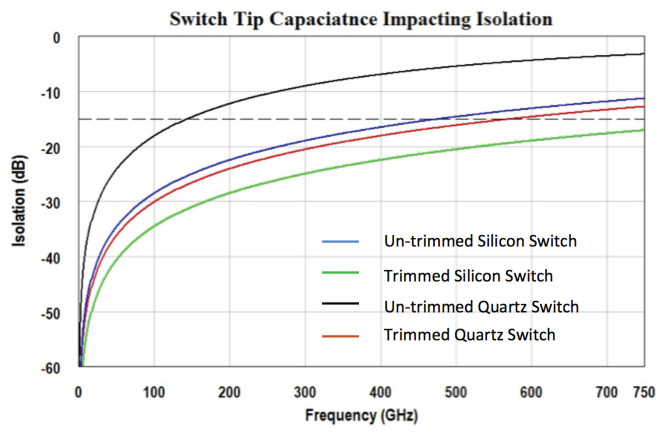
Estimated switch isolation performance dominated by actuator tip capacitance.

**Figure 9 micromachines-13-00745-f009:**
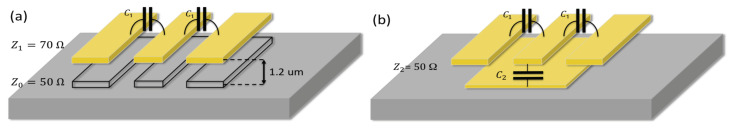
(**a**) Impedance variation caused by CPW structure elevation. (**b**) impedance tuning is realized by adjusting the DC bias pad size.

**Figure 10 micromachines-13-00745-f010:**
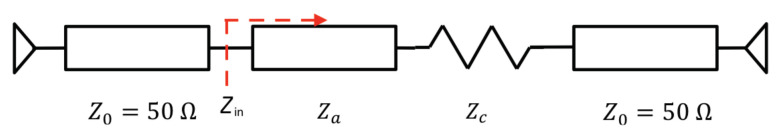
Simplified transmission line model of MEMS switch at the ON state.

**Figure 11 micromachines-13-00745-f011:**
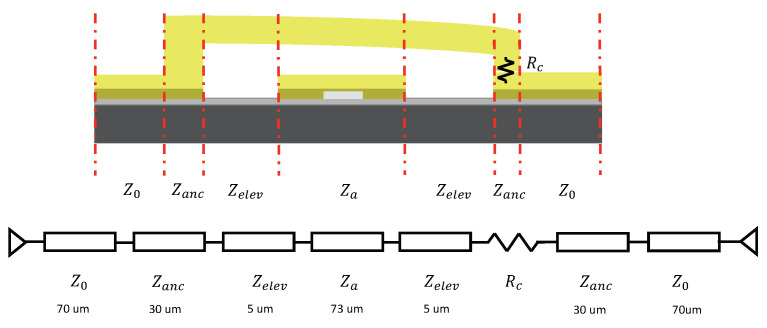
A comprehensive transmission line circuit model for ON state MEMS switch.

**Figure 12 micromachines-13-00745-f012:**
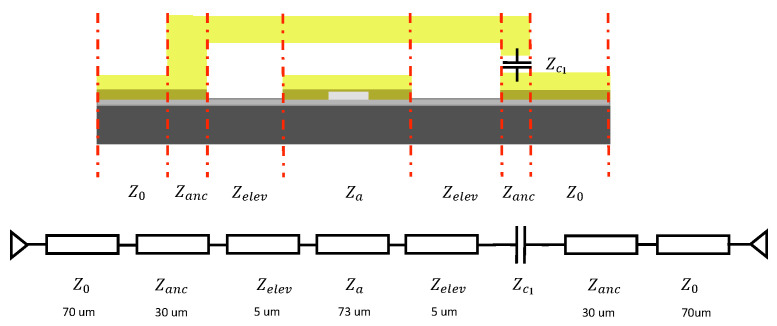
A comprehensive transmission line circuit model for OFF state MEMS switch.

**Figure 13 micromachines-13-00745-f013:**
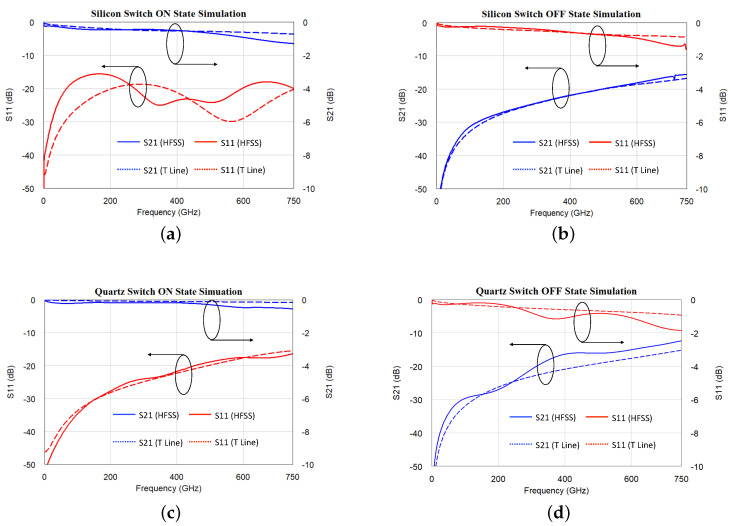
The THz MEMS switches were modeled as transmission line circuits and simulated using AWR Microwave Office (marked as T line). The circuit simulation results are compared with ANSYS HFSS finite element analysis. (**a**) Silicon based design at the ON state. (**b**) Silicon based design at the OFF state. (**c**) Quartz based design at the ON state. (**d**) Quartz based design at the OFF state.

**Figure 14 micromachines-13-00745-f014:**
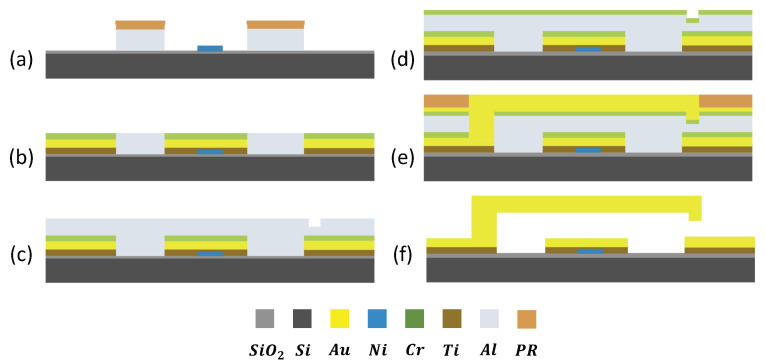
A simplified silicon-based MEMS switch fabrication flow. The fabrication process starts with circuit layer deposition using a lift-off technique in (**a**,**b**). After that, two aluminum sacrificial layers are deposited and dimple position is prepared with another lift-off in (**c**). Etching and plating were used to form the switch’s anchor in (**d**). After the gold seed layer and a photoresist patterning, the beam was plated in (**e**). After a series of wet etch steps, the switch is finally released by CPD in (**f**).

**Figure 15 micromachines-13-00745-f015:**
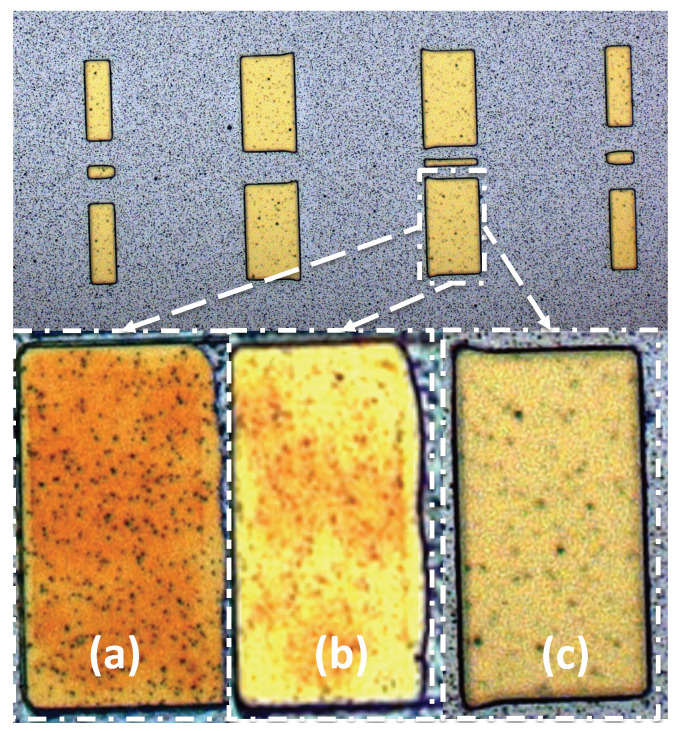
(**a**) The Al-Au compounds observed (**b**) Al-Au compounds are reduced with 50 nm chromium barrier layer applied between the gold and aluminum layer (**c**) 80 nm chromium barrier layer prevented Al-Au to produce.

**Figure 16 micromachines-13-00745-f016:**
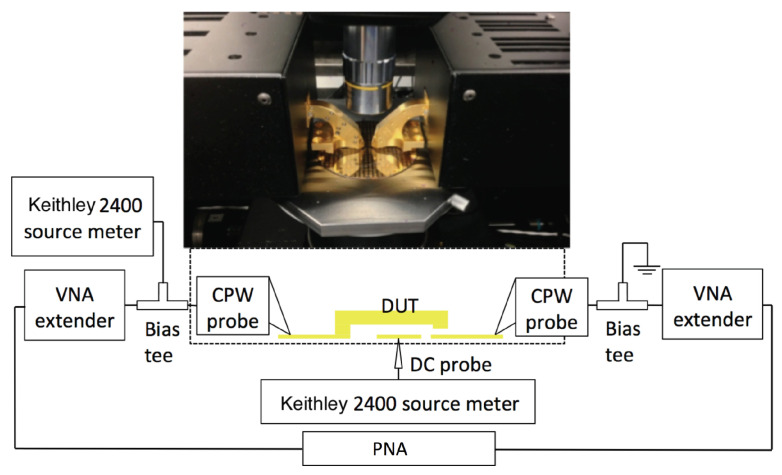
On wafter two-ports probing set up used for MEMS switch RF measurement is shown in this diagram.

**Figure 17 micromachines-13-00745-f017:**
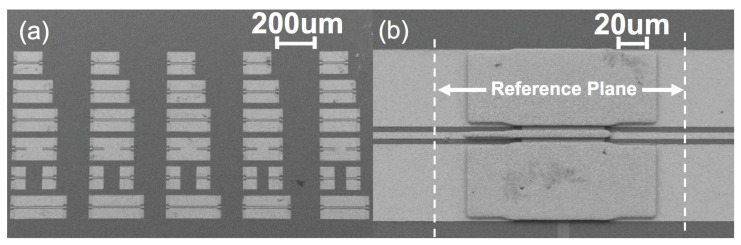
SEM image of (**a**) on-wafer TRL calibration kit and (**b**) an example silicon switch.

**Figure 18 micromachines-13-00745-f018:**
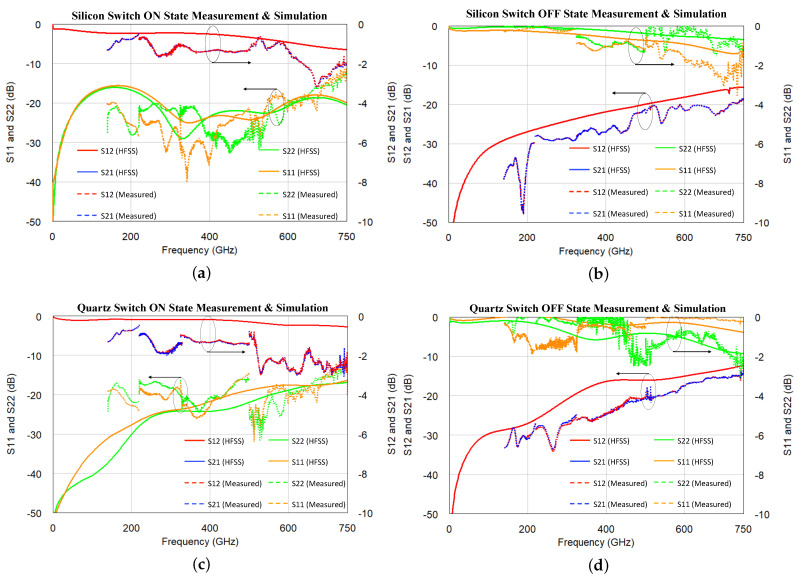
The THz MEMS switch HFSS simulation measurement comparison. (**a**) Silicon switch ON state measurement and simulation. (**b**) Silicon switch OFF state measurement and simulation. (**c**) Quartz switch ON state measurement and simulation. (**d**) Quartz switch OFF state measurement and simulation.

**Table 1 micromachines-13-00745-t001:** The cross-section dimensions of a 50 Ω CPW on different substrates.

Substrate Material	Relative Permittivity ϵ	*g* (μm)	*w* (μm)
Silicon	11.9	4	7
SiC	9.7	4	9
AlN	9.2	4	10
Quartz	4.0	4	35

The dimensions of 50 Ω CPWs on the different substrates are compared through Ansys HFSS simulation. In the CPW design, the signal-ground gap *g*_c_ is limited to 4 μm to keep the transmission line fabrication feasible. With the same *g*_c_, the differences in signal line width *w* are compared under different relative permittivity *ϵ*.

**Table 2 micromachines-13-00745-t002:** CPW impedance & its elevation height above the silicon substrate.

Elevation Height (μm)	Impedance (Ω)
0	50
0.4	63.6
0.8	64.3
1.2	70

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
