# Peer review of "THz MEMS Switch Design"

_micromachines, 2022, doi:10.3390/mi13050745_

Round 1
Reviewer 2 Report
Comments to the Author
This manuscript contains very interesting ideas for improving RF performance of MEMS switches in order to achieve THz performance. However, I recommend resolving the following issues before publishing.
1) The dielectric strength of fused quartz is mentioned to be 10MV/cm in line 59, whereas it is reported as 0.2-0.4 MV/cm. It is not clear what is the difference between these two scenarios. Please clarify.
2) Line 73 mentions that a reliable restoring force would be around 0.07 mN, as reported in a previous work. The reliable restoring force would depend on the adhesion force at the contact, i.e. the contact area and the contact material. Please confirm that these design parameters are the same as the cited work.
3) The pull-down voltage in equation (2) mentions the parameter g as the gap between the cantilever and the actuator, which is the actuation gap of the device. However, in Fig. 1(a) g is shown as the contact gap, i.e. the gap between the cantilever tip and contact pad. The fabrication flow in Fig 14 shows that these two gaps are not the same. Please clarify and mention the values of these two gaps. Also please note that equation (2) works if the contact gap is > one-third of the actuation gap. If this relationship is not true, the voltage equation has to be changed to the expression for “non-pull-in mode”.
4) Line 69 shows W as the overlapping between the cantilever and conductor beneath it. Please identify this parameter in Fig. 1(a).
5) Fig. 9 is referenced just after Fig. 4. Please change the number of this figure to 5 for ease of reading. Also please fix the caption of Fig. 3.
6) In line 122, the tip capacitance for silicon and fused quartz substrates are reported as 0.6 fF and 2fF. It is mentioned that contact overlaps are the same for both substrates. Since Table 1 shows the gap to be the same for both substrates and the width for quartz switch to be 5 times the width for silicon substrate, ideally the capacitance for quartz switch should be 5 times the capacitance for silicon switch. Please explain this discrepancy. Does the equation used here include the parasitic capacitance?
7) Line 128 mentions trimming of the cantilever tip. Please refer to Fig. 6 (b) and Fig. 7(b) in this context (These figures are not referenced anywhere). Fig. 8 shows results for trimmed and un-trimmed switches. So please explain the idea of trimming before referring to Fig. 8.
8) Fig. 9 and the corresponding text mentions the structure elevation. It is not clear to the reviewer whether the authors mean electrostatic levitation or otherwise. Are the fringing field effects and capacitances to the substrate relevant here? Please re-write the text associated with this part to explain.
9) The reviewer suggests revising the manuscript to correct grammatical errors. For example, lines 21/30/ 38 on the first page need to revised.
10) The authors’ previous work verified RF switch operation at frequencies up to 750GHz. In this work, the experimental results are verified at the same maximum frequency, however, the proposed design is supposed to operate at 1THz. It would be helpful to provide a succinct description of the major improvements over the previous designs.
Round 2
Reviewer 1 Report
In the response letter, the authors have answered all of my comments.
Therefore, I recommend for publication.
Reviewer 2 Report
Comments to the Author
Thanks to the authors for addressing most of the comments/ concerns from this reviewer carefully. The reviewer would like to recommend a few minor changes before publishing:
- The new version of equation 2 shows the expression for the pull-in voltage "Vpull", where the “Lw” part in the denominator of this equation should represent the actuation area. Hence L should be the overlapping length between the bias (actuation) pad and the cantilever. However, Fig. 1 shows L as the overlap between the cantilever and the CPW (contact pad). Please correct this. It might be useful to show the parameter L in Fig. 2 as well. Would this change affect the reported initial value of Vpull?
- In the four plots in Fig. 14 (a)-(d), S12/S21 have a widely different range than S11/S22. Hence all four curves cannot be viewed in their full range in order to correlate with the observations made by the authors in Section 3. The reviewer recommends using two different y-axes (similar to Fig. 19) for S12/S21 and S11/S22.
- The reviewer recommends checking the whole manuscript rigorously for grammatical errors (e.g. mixing of present and past tense in the same sentence in the literature review) or structural errors (e.g. Line 187). The reviewer also recommends avoiding contractions like “can’t”/“won’t” and opting for full forms instead, which is standard for scientific literature.
- Fig. 18 is referenced before Fig. 17. Please swap their positions.
- Please make the plots in Fig. 19 is bolder lines for better visibility.
